# Monitoring the Impacts of Human Activities on Urban Ecosystems Based on the Enhanced UCCLN (EUCCLN) Model

Nadia Abbaszadeh Tehrani *, Farinaz Farhanj [ID] and Milad Janalipour [ID]

Assistant Professor, Aerospace Research Institute, Ministry of Science, Research, and Technology, Tehran 1465774111, Iran; milad_janalipour@ari.ac.ir (M.J.)
* Correspondence: tehrani@ari.ac.ir; Tel.: +98-21-88366035-38

**Abstract:** To have a sustainable city, human pressures on urban ecosystems should not exceed certain thresholds, which are defined by the urban carrying capacity concept. The main goal of this research was to monitor environmental pressures caused by the impacts of human activities on the ecosystem of Tehran city using spatial indicators. According to the enhanced Urban Carrying Capacity Load Number (EUCCLN) model, first, the most related indicators were collected from the open access databases, including satellite products, air quality monitoring stations, municipality statistical yearbook, and a related article. Then, the indicators were classified into air, traffic, and waste groups. Afterwards, the importance coefficients of all indicators were specified using the analytical hierarchy process. Their degree of carrying capacity tables were determined, and finally, load numbers were calculated. The results showed that 100%, 4.55%, and 40.91% of all districts had very high-to-critical degrees in terms of air, traffic, and waste indicators, respectively. The final human-induced pressure degrees were very high-to-critical in Districts 1, 3, 6, 7, 8, 12, and 14 (31.82% out of 22 districts) and high-to-very high in the rest of them. Therefore, the overall pressure in all 22 districts of Tehran had reached or exceeded its maximum threshold degree.

**Keywords:** enhanced urban carrying capacity load number (EUCCLN) model; environmental impacts of human activities; geographic information system; Google Earth Engine; load number; remote sensing; urban carrying capacity; urban ecosystem

## 1. Introduction

In the recent decades, the planet has experienced an unprecedented increasing rate of urbanization. This growth has occurred due to the migration of a large number of rural people to cities [1]. This high speed of urban development, along with the increasing population growth rate, has caused problems such as the increase in air pollution, traffic jams, waste production, natural resources destruction, crowding, water pollution, land use changes, lack of public urban facilities, disruption of urban metabolism as a result of excessive consumption of materials and energy, etc. [2,3]. These mentioned problems have increased environmental pressures caused by the impacts of human activities on urban ecosystems, which are considered a serious threat to their resilience and sustainability [4]. The human-induced pressures should not exceed the defined thresholds, which are referred to as carrying capacity [5]. Therefore, when adopting policies related to urbanization, it is necessary to pay attention to the Urban Carrying Capacity (UCC) concepts [6]. Carrying capacity was initially proposed in the 1890s by some managers who paid special attention to the use of rangeland for livestock and wildlife grazing [1,7]. Nowadays, UCC refers to the level of human activities, land use changes, physical developments, population growth, per capita material and energy consumptions, and waste production that can be sustainably supported in an area, without causing irreversible damages [8,9]. Environmental carrying capacity also refers to the level of human activities in the urban area in which the natural environment can absorb the wastes and pollutants produced, and provides

them with sufficient natural resources and sinks [10]. Therefore, the carrying capacity from an environmental point of view is related to the two criteria of "source" and "sink", or the capacity to supply natural resources and absorb waste and pollutants [1,7,11]. The UCC includes the fundamental concepts of (1) entropy rate or the rate of absorption of pollution and waste in the urban ecosystem, (2) the capacity of source-sink, and (3) the analysis of urban metabolism. Urban sustainability also includes two main rules: optimal access to facilities, and justice in the distribution of environmental resources, benefits, and harms [12]. The United Nations Environment Program (UNEP) has suggested the Pressure–State–Impact–Response (PSIR) framework to assess environmental pressures, ecosystem sustainability, and urban carrying capacity, and generally prepare environmental status reports. PSIR is a systematic framework of environmental information under four groups: pressures or factors of environmental changes, environmental status, impacts of human activities, and human policies to reduce the intensity of these pressures [13]. UCC monitoring indicators have been selected based on the PSIR framework and the concepts of UCC and urban sustainability.

To investigate the environmental state of the urban ecosystem, some previous studies have used descriptive–analytical approaches [14–19]. While, others have utilized numerical models for considering the concepts of UCC, such as Urban Carrying Capacity Load Number (UCCLN) [5], a combination of Geographic Information System Fuzzy Modeling (GISFM) and Technique for Order of Preference by Similarity to Ideal Solution (TOPSIS) [20], sum-weighted regulations [21], Weighted-Aggregated Sum Product Assessment (WAS-PAS) [22], and Urban Carrying Capacity Coupling Model (UCCCM) [23]. Abbaszadeh Tehrani and Makhdoum [5] monitored the UCC of the Tehran metropolis using groups of indicators of natural condition, population, energy, water, air, traffic, waste, and land use based on the UCCLN model. They showed that environmental pressures were critical in 35% of the districts, and pressures of water, energy, population, and waste were higher than other indicators. The advantages of this model are: (1) its high efficiency in managing man-made ecosystems, (2) high flexibility in estimating the UCC with high precision in a completely localized manner, (3) considering the desirable, acceptable, and critical thresholds of indicators, and (4) high efficiency in organizing them [4,24,25]. Irankhahi et al. [20] estimated the UCC of Shemiran city in the Tehran province using the combination of GISFM and TOPSIS. They concluded that none of the districts had desirable degrees of environmental pressure. Azizi [21] used sum-weighted regulations to evaluate the UCC in Tehran city. It is concluded that the environmental capacity was weaker in the central and southern parts of the city, including Narmak, Daryan Nou, Sizdah-Aban, and Nazi Abad. In the research, the stability threshold values of each index were not determined. Esfandi and Nourian [22] also estimated the UCC of Tehran's districts by considering the groups of economic, social, environmental, and traffic indicators using the WASPAS model. They concluded that the UCC was weak and very weak in Districts 1, 3, 6, and 7 to 20. Cao et al. [23] assessed the UCC of Shanghai in China using the UCCCM model. They concluded that the UCC was more critical in urban areas that were exposed to pollutants from heavy industries and had older fabrics. The limitation of their research was the lack of determination of stability threshold values for each of the indicators.

The main purpose of this research was to use an enhanced UCCLN (EUCCLN) model to monitor the environmental pressures caused by the negative impacts of human activities (the third part of the PSIR framework) on the urban ecosystem of Tehran city. Additionally, the specific objectives of this research were to estimate the distance of the spatial indicators of human impacts with desirable, acceptable, threshold, and critical values based on the concepts of the UCC and the integration of remote sensing and Geographic Information System (GIS) for better monitoring of urban ecosystems. The innovation of this paper was in improving the initial UCCLN model by using remote sensing data to provide more up-to-date and effective human-induced pressure indicators. Google Earth Engine (GEE), as a powerful platform, is used for gathering and analyzing spatial data and information.

## 2. Materials and Methods

### 2.1. Study Area

The study area in this research was Tehran city in the north of Tehran province and in the central north of Iran (Figure 1). This city is divided into 22 districts, which is one of the largest cities in the world with an area of 751 km$^2$. It is located at the latitude 35°43′ N and longitude 51°24′ E. This city is limited from the south, north, east, and west sides to Varamin city, the southern Alborz mountain range, Lavasan city, and Karaj city, respectively [26]. Its minimum and maximum heights are 900 and 1800 m, respectively. According to the 2016 census, approximately 8.694 million people live in this city [27]. Tehran has been involved in lots of environmental problems, such as the increase in air pollution, traffic jams, and the excessive amount of waste production, due to the unbalanced growth of its population, in the last few decades.

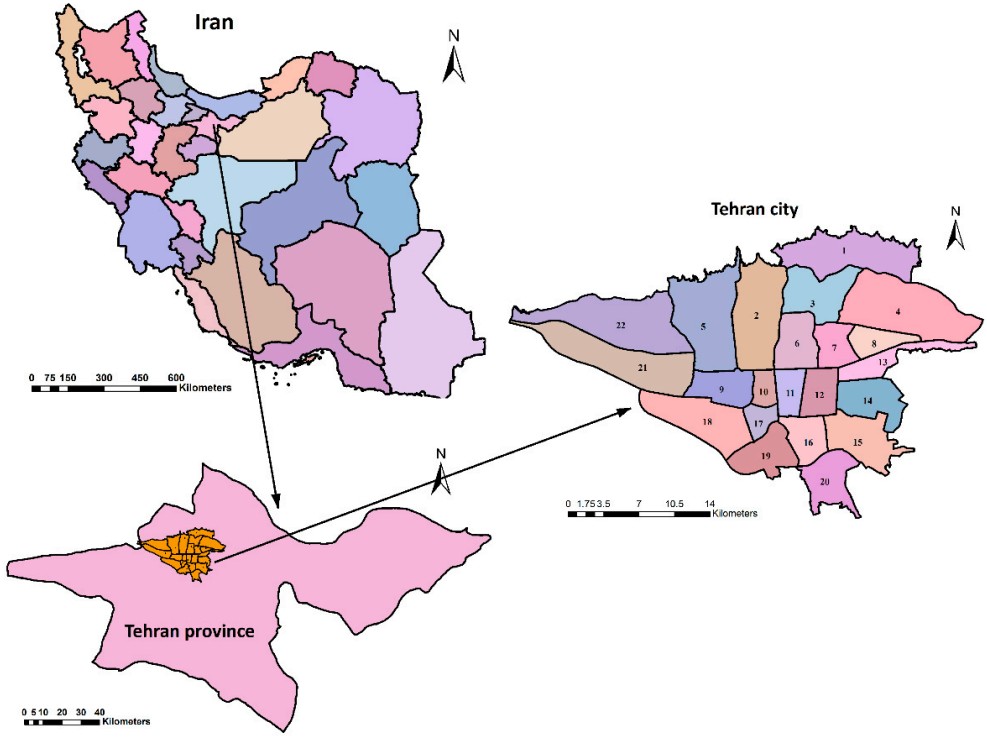

**Figure 1.** The geographic location of the study area (Tehran city, Iran).

### 2.2. Data

To estimate the environmental pressures caused by human impacts using the EUCCLN model, three components of air (including five indicators of the average concentration of nitrogen dioxide (NO$_2$), sulfur dioxide (SO$_2$), particulate matter smaller than 2.5 μm (PM$_{2.5}$), particulate matter smaller than 10 μm (PM$_{10}$), and ozone (O$_3$)), traffic (including two indicators of intra-city trip generation rate, and intra-city trip attraction rate), and waste (including three indicators of waste production, waste production rate, and recycling rate) were chosen. It should be noted that the differences between the indicators considered in the EUCCLN model and its initial version [5] were in replacing the average concentration of NO$_2$, SO$_2$, PM$_{2.5}$, PM$_{10}$, and O$_3$ pollutants instead of the average carbon monoxide (CO) emission, and the number of polluted days of a year in the air component, as well as the replacement of intra-city trip generation and attraction rate indicators instead of the traffic volume indicator in the traffic component. According to [28], only 17 clean days were reported in 2020 in Tehran. The fact that the number of polluted days in this mentioned year was much higher than the maximum acceptable threshold, which is equal to 5% per year [5], showed that there was a need to use more spatial, monitorable, and dynamic air indicators in the enhanced model. In addition, nowadays, the concentration of

atmospheric pollutants is easily available using new technologies, such as remote sensing sensors, with high accuracy in an appropriate spatial resolution. In the traffic component, the replacement of the mentioned indicators was due to the high importance of intra-city trip rate in the UCC modeling [22].

Sentinel 5-P open access satellite products were used to obtain the average concentration of $NO_2$ and $SO_2$ [29]. Sentinel 5-P is one of the Earth observation satellites that was launched in October 2017. The sensor used in this satellite is the tropospheric monitoring instrument (TROPOMI), which is designed to monitor the atmosphere, weather, and air quality at a spatial resolution of approximately 1 km [30]. The average values of the mentioned pollutants in mol/m$^2$ unit were obtained from the GEE in 2020 for the city of Tehran in GeoTIFF format. GEE is a high-performance open-source cloud computing system for storing, processing, visualizing, and analyzing time series of geospatial data and remote sensing products [31]. It should be noted that "NO2_column_number_density" and "SO2_column_number_density" bands were used in this regard. Some researchers have concluded in their studies that the accuracy of Sentinel 5-P in monitoring the mentioned pollutants ranges from 0.5 to 0.81 [32–35]. Veefkind et al. [36] have also indicated that the accuracy and reliability of Sentinel 5-P in monitoring pollutants is acceptable and meets the needs. The average concentration of $PM_{2.5}$, $PM_{10}$, and $O_3$ values in each of the air quality monitoring stations in 2020 was obtained from [37]. It should be noted that data from these monitoring stations were freely available [37]. The hourly average values of $O_3$ in a ppb unit on each day from 20 March 2020 to 20 September 2020, were obtained from each station, and the average of its 8 h values was calculated. In the following, maximum values of these 8 h averages were selected, and, finally, the mean $O_3$ concentration values in each station over a period of the mentioned six months were computed. After collecting the data, $PM_{2.5}$, $PM_{10}$, and $O_3$ were interpolated to a spatial resolution of 1 km using the kriging method.

Intra-city trip generation and attraction rate (number of trip generations and attraction per 100 m$^2$) of Tehran were obtained from the municipality statistical yearbook [38] in 2020 in 22 districts that were available as open access.

To calculate the indicators related to the waste criterion, the total amounts of waste in 2020 and 2019 were obtained from the municipality statistical yearbook that was openly available [38,39]. The waste production index in ton/ha unit was provided by dividing the total amounts of waste in 2020 [38] by the area of districts. The waste production rate index in 2020 compared to 2019 was calculated based on the total amount of dry and wet waste collected in each of the 22 districts of Tehran during these two years [38,39], by applying the following equation:

$$W_n = W_0 (1 + r)^n, \tag{1}$$

where, $W_n$ and $W_0$ are the amounts of waste production in 2020 and 2019, respectively, n is equal to 1 (one-year difference between 2018 and 2019), and r is the waste production rate in each district. The recycling rate in the last decade in Tehran has been almost constant and equal to 18% [40]. Due to the unavailability of recycling rate values in each of the 22 districts, this index was considered equal to 18% in all of them. It should be noted that all traffic and waste indicators were converted to a spatial resolution of 1 km. Indicators selected to monitor the UCC of the impacts of human activities were summarized in Table 1.

**Table 1.** Indicators for monitoring the environmental pressures caused by the impacts of human activities on the ecosystem of Tehran.

| Component | Indicator | | Measuring Unit | Data Source |
|---|---|---|---|---|
| air | average concentration of | $NO_2$ $SO_2$ | $mol/m^2$ | [29] |
| | | $PM_{2.5}$ $PM_{10}$ | $\mu g/m^3$ | [37] |
| | | $O_3$ | ppb | |
| traffic | intra-city trip | generation rate attraction rate | number of trips per 100 m$^2$ | [38] |
| waste | waste production waste production rate recycling rate | | ton/ha % | [38,39] [40] |

## *2.3. Methodology*

In general, the UCCLN model has been developed using a total of 35 indicators based on the PSIR framework in the criteria of pressure consisting of gas, electricity, and water indicators, states including natural and man-made state indicators, and impact including air, traffic, and waste indicators. In this research, as mentioned, the third component of PSIR, i.e., impacts consisting of 10 indicators of the average concentration of $NO_2$, $SO_2$, $PM_{2.5}$, $PM_{10}$, and $O_3$, intra-city trip generation and attraction rate, waste production, waste production rate, and waste recycling rate were investigated. The research method flowchart adopted to estimate the environmental pressures using the EUCCLN model is shown in Figure 2. In the following, the description of each step has been discussed.

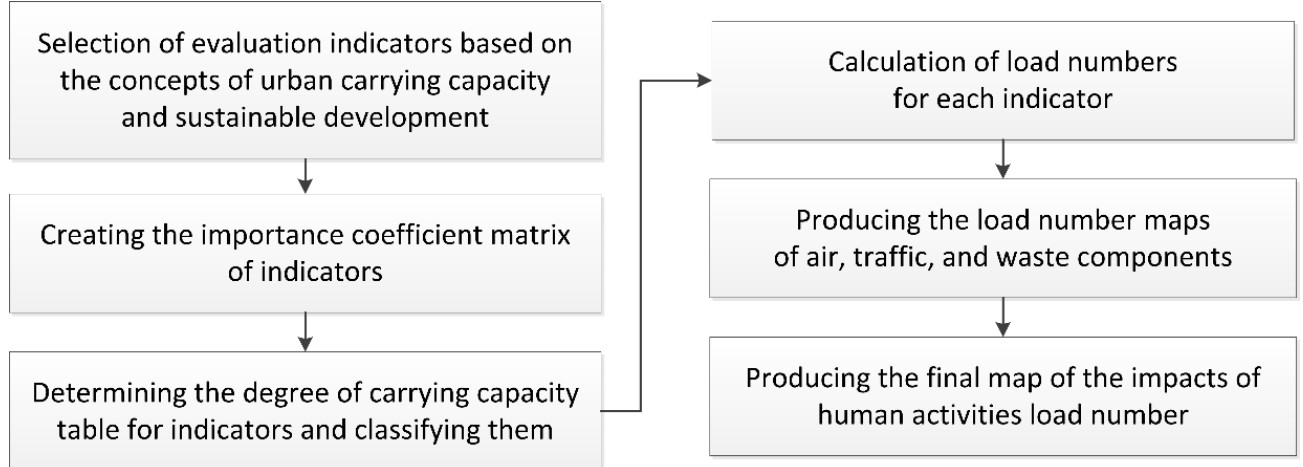

**Figure 2.** The research method flowchart.

- **Selection of the evaluation indicators**:

Relying on the concepts of UCC and sustainability, the three components of air, traffic, and waste consist of 10 indicators of the average concentration of $NO_2$, $SO_2$, $PM_{2.5}$, $PM_{10}$, and $O_3$, intra-city trip generation and attraction rate, waste production, waste production rate, and recycling rate were selected to monitor the environmental pressures caused by impacts of human activities on the Tehran ecosystem.

- **Creating the importance coefficient matrix of indicators**:

Analytical hierarchy process (AHP) [41], which is one of the most popular methods of multi-criteria evaluation (MCE), was used to determine the importance coefficient (IC) of each of the 10 indicators in estimating the UCC of impacts. AHP is a method for making

decisions in complex problems that include a large number of indicators and the importance of each of them needs to be determined. The weights of indicators were carried out by three experts in environmental and urban sciences in the Expert Choice software. The experts were familiar with the environmental problems of Tehran's ecosystem and the concepts of UCC and sustainability. Moreover, the determined weights were confirmed by experts of the Tehran municipality. The structure of our decision-making problem was designed in two levels, including criteria and sub-criteria. Four questionnaires based on Saaty's pairwise comparison table were employed. The first questionnaire is regarding the criteria, i.e., air, traffic, and waste. The second, third, and fourth questionnaires are used for the sub-criteria, i.e., $NO_2$, $SO_2$, $PM_{2.5}$, $PM_{10}$, $O_3$; generation rate, attraction rate; and waste production, waste production rate, and recycling rate.

- **Determining the degree of carrying capacity indicators**:

Each indicator was classified into six pressure classes including: very low, low, medium, high, very high, and critical. Based on the amount of each indicator in the range of its carrying capacity, a grade was assigned to the classes, which is called the Degree of Carrying Capacity (DCC). The six classes of DCC and their meanings are presented in Table 2 [25].

**Table 2.** DCC classes and their meanings.

| Class | 1 | 2 | 3 | 4 | 5 | 6 |
|---|---|---|---|---|---|---|
| **DCC** | 0.1 | 1 | 2 | 3 | 4 | 5 |
| pressure degree | very low | low | medium | high | very high | critical (exceeding the threshold) |
| meaning of DCC | desirable degree | degrees of pressure increasing from low to high | | | threshold degree | critical degree |

The DCC of the average concentration of $NO_2$ and $SO_2$ were considered based on equal intervals of their values [22] from the minimum to the maximum ones. The DCC of $PM_{2.5}$, $PM_{10}$, and $O_3$ were set according to the existing international standards provided by the World Health Organization (WHO) [42]. It should be noted that standards provided for the $PM_{2.5}$ and $PM_{10}$ were based on their yearly averages. Moreover, the standard provided for $O_3$ was based on the average daily maximum 8 h mean value in the peak seasons. According to the definition of the WHO, the peak seasons for $O_3$ were six consecutive months of the year when the highest emissions of this pollutant were expected. For countries located in the northern hemisphere, such as Iran, this period should be the warm seasons of the year, i.e., spring and summer [42]. The DCC of the traffic indicators, i.e., trip generation and attraction rate were also included according to equal intervals [22] based on their values. The maximum allowed production of waste, based on the number of reservoirs available for disposal or recycling, is equal to 0.5 kg per day per person, or 182.5 kg per year per person [12]. The optimal level of the population density in cities, which is equal to 50 persons/ha, was used to obtain the desired level of waste production ($182.5 \times 50 = 9125$ kg/ha) [4,25]. Thus, the desirable level of waste production is equal to 9125 kg/ha, or approximately 9 tons/ha ($9125/1000 \cong 9$) [4,12,20,25]. Based on the available facilities, it is not possible to dispose of more than 8000 tons of waste per day, or 2,920,000 tons per year in Tehran [20]. Since the area of this city is equal to 61,562 ha [26], the threshold or the maximum acceptable level of waste production is equal to 47 tons/ha ($2,920,000/61,562 \cong 47$). Since the waste production and its rate were different in the 22 districts of Tehran, the desirable and the maximum acceptable levels of waste production rate were also different in each district. To determine the DCC table of waste production rate, the values of this indicator until reaching the desired, and accepted level of waste production (9 tons/ha, and 47 tons/ha, respectively), in a period of 25 years were calculated using Equation (1) for each of the 22 districts separately [20]. In this expression,

in each district, by setting $W_n = 47$, the maximum acceptable waste production rate, and once again by setting $W_n = 9$, the desirable rate were obtained. $W_0$ was also equal to the value of waste production in 2020 (tons/ha), and n was equal to 25. For determining DCC classes of the recycling rate indicator, recycling rates of more than 80% and less than 20% were considered desirable and the maximum acceptable pressure level, respectively [12,25].

- **Calculation of Load Numbers (LN) for each indicator**:

After classifying indicators based on the relevant DCC tables, the IC of each indicator was multiplied by its DCC value to calculate the Load Numbers (LN), which is presented as follows [4,25]:

$$LN = IC \times DCC \tag{2}$$

LN represents the environmental pressure on the urban ecosystem [4,25].

- **Producing the LN maps of components**:

The LN maps of the air, traffic, and waste components were produced by the total load numbers of their relevant indicators in each group.

- **Producing the final LN map of the impacts of human activities**:

By summing all load numbers of the air, traffic, and waste components in each of the 1 km cells, the final LN map of human impacts on the ecosystem of Tehran was obtained.

## 3. Results

### 3.1. Importance Coefficient (IC) of Indicators

The values of the IC of indicators obtained using the AHP method are displayed in order of priority from the highest importance to the lowest in Figure 3. It should be noted that the amount of inconsistency calculated to determine the weights by the AHP method was equal to 0.00021, and therefore the matrix of pairwise comparisons of indicators was considered compatible. From Figure 3, it can be concluded that the average concentration of $PM_{2.5}$ (IC = 11.5), $NO_2$, $SO_2$, and $PM_{10}$ (IC = 10.7), trip attraction rate (IC = 10.6), waste production rate (IC = 9.6), $O_3$ and trip generation rate (IC = 9.5), waste production (IC = 8.8), and recycling rate (IC = 8.5) were the most important in monitoring the UCC, respectively.

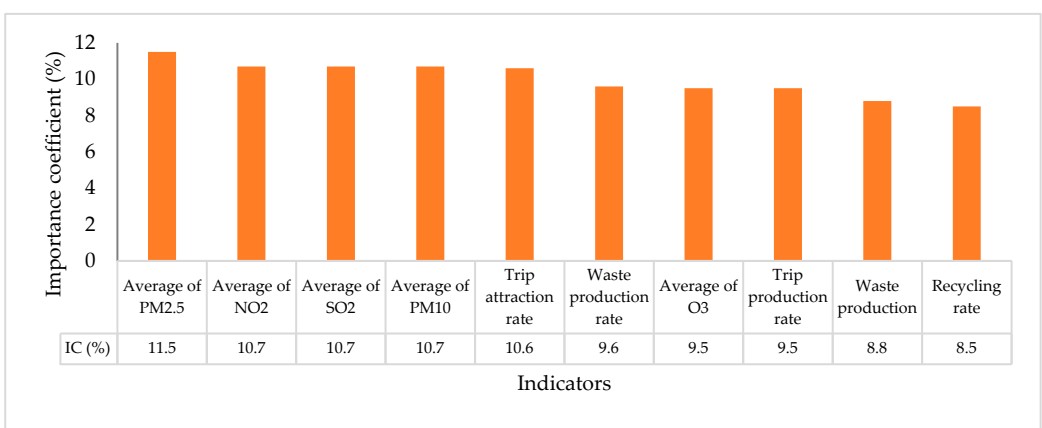

**Figure 3.** The importance coefficient of indicators for monitoring the UCC of human impacts.

### 3.2. The DCC Table of Indicators

The DCC classes of all indicators are provided in Table 3. Based on the ranges presented in this table, all indicators were classified into six pressure groups from desirable to critical degrees.

**Table 3.** DCC classes of indicators.

| Indicator | | Measuring Unit | DCC (Pressure Degree) | | | | | |
|---|---|---|---|---|---|---|---|---|
| | | | 0.1 (Desirable) | 1 (Low) | 2 (Medium) | 3 (High) | 4 (Threshold) | 5 (Critical) |
| average concentration of | $NO_2$ | $\mu mol/m^2$ | 0–160.5 | 160.6–321 | 321.1–481.5 | 481.6–642 | 642.1–802.5 | 802.6–963 |
| | $SO_2$ | | 0–65.5 | 65.6–131 | 131.1–196.5 | 196.6–262 | 262.1–327.5 | 327.6–393 |
| | $PM_{2.5}$ | $\mu g/m^3$ | 0–5 | 5.01–10 | 10.01–15 | 15.01–20 | 20.01–25 | 25< |
| | $PM_{10}$ | | 0–15 | 15.01–20 | 20.01–30 | 30.01–40 | 40.01–50 | 50< |
| | $O_3$ | ppb | 0–15 | 15.01–31 | 31.01–32 | 32.01–33 | 33.01–36 | 36.01–51 |
| intra-city trip | generation rate | number of trips per 100 m$^2$ | 0–0.2–5 | 0.26–0.5 | 0.51–0.75 | 0.76–1 | 1.01–1.25 | 1.26–1.5 |
| | attraction rate | | 0–0.5 | 0.51–1 | 1.01–1.5 | 1.51–2 | 2.01–2.5 | 2.51–3.1 |
| waste production | | ton/ha | 0–9 | 9.01–19 | 19.01–29 | 29.01–38 | 38.01–47 | 47< |
| waste production rate | | % | | | | * | | |
| recycling rate | | | 80–100 | 65–79.99 | 50–64.99 | 35–49.99 | 20–34.99 | <20 |

* The desirable and maximum acceptable levels were calculated for each of the 22 districts separately, using Equation (1).

### 3.3. DCC and LN Maps of Indicators

The DCC and LN maps of the average concentration of $NO_2$, $SO_2$, $PM_{2.5}$, $PM_{10}$, and $O_3$, at a spatial resolution of 1 km in Tehran, are shown in Figure 4a–e, respectively. According to Figure 4a, in terms of $NO_2$, the majority parts of Districts 3, 6 to 8, and 11 to 13 had the highest pressure, equivalent to the critical level, and the majority parts of the southern, southwestern, and western Districts 18 to 22 had the lowest pressure, equivalent to the high degree among other districts. According to Figure 4b, in terms of $SO_2$, the highest pressure, equivalent to the critical degree, was observed in most parts of southern Districts (15 and 20), and the lowest pressure, equivalent to the medium degree, was observed in small parts of Districts 2, 4, 5, and 22. It can be concluded that most of the environmental pressure degrees caused by $NO_2$ and $SO_2$ were high, very high, and critical. According to Figure 4c–e, based on $PM_{2.5}$, $PM_{10}$, and $O_3$, all districts had a critical pressure degree that exceeded the maximum acceptable threshold level.

The DCC and LN maps of intra-city trip generation and attraction rate indicators are presented in Figure 5a,b, respectively. According to Figure 5a, in terms of the trip generation rate, the highest pressure level observed in District 6 was equivalent to the critical level, and the lowest level seen in Districts 15, 16, and 18 to 20, was equivalent to the medium degree. Based on Figure 5b (trip attraction rate), the highest pressure, or critical degree, was observed in District 6, and the lowest or very low (desirable) degree was seen in Districts 10, 14, 15, and 17.

The DCC and LN maps of waste indicators, including waste production, waste production rate, and recycling rate, are shown in Figure 6a–c, respectively. According to Figure 6a (waste production), the highest pressure, which is equivalent to the critical degree, was in Districts 7, 8, 10 to 12, 14, 15, and 17, and the lowest one, or very low degree, was observed in the western Districts (21 and 22). Based on Figure 6b (waste production rate in 2020 compared to 2019), Districts 1, 3, 6, 8, 10, 14, and 17 had the highest pressure or critical degree, and Districts 2, 4, 5, 11, 12, 21, and 22 had the lowest pressure or very low degree. According to Figure 6c (recycling rate), the pressure in all 22 districts was equivalent to the critical level.

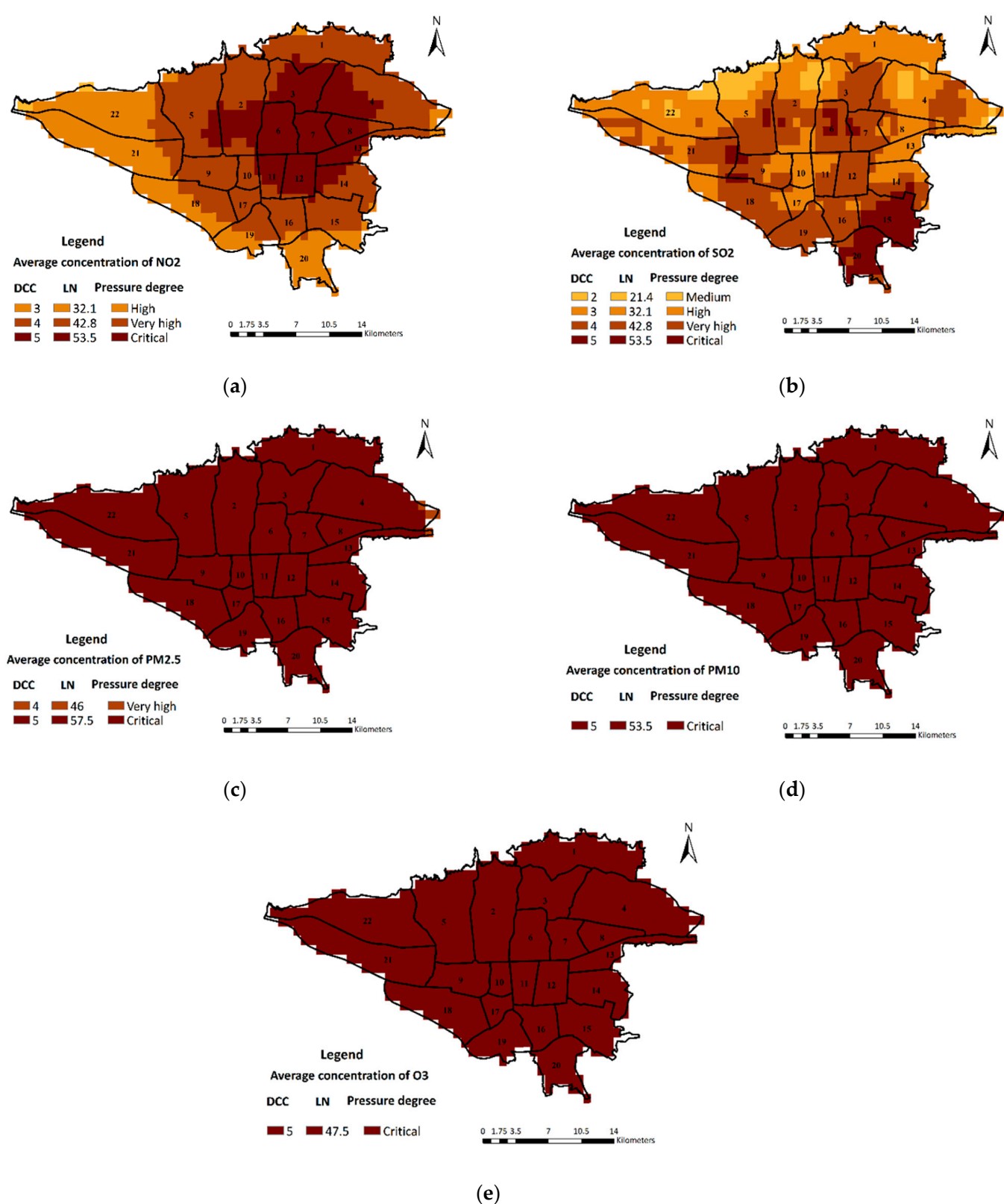

**Figure 4.** The DCC and LN maps of air indicators. (**a**) NO$_2$; (**b**) SO$_2$; (**c**) PM$_{2.5}$; (**d**) PM$_{10}$; (**e**) O$_3$.

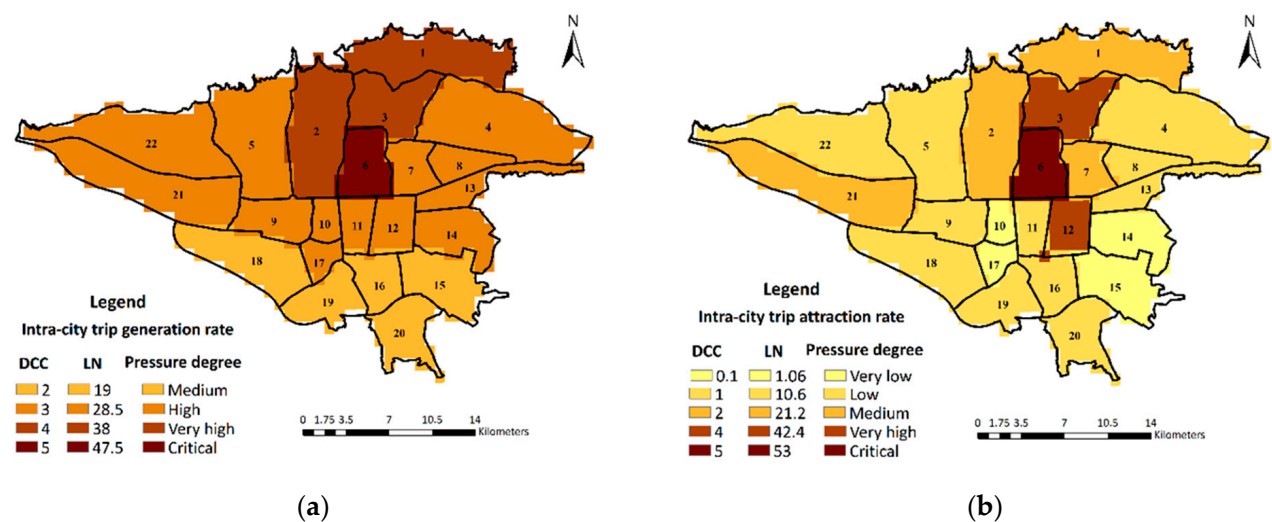

**Figure 5.** The DCC and LN maps of traffic indicators. (**a**) Trip generation rate; (**b**) trip attraction rate.

**Figure 6.** The DCC and LN maps of waste indicators. (**a**) Waste production; (**b**) waste production rate; (**c**) recycling rate.

*3.4. LN Maps of Components*

The LN maps of impact components (air, traffic, and waste indicators) in Tehran, at a spatial resolution of 1 km, are presented in Figure 7a–c, respectively. According to Figure 7a, it can be concluded that almost all 22 districts had very high-to-critical pressure levels in the air component, equivalent to the critical UCC degree. Based on Figure 7b, traffic had the highest pressure on the environment of District 6, equivalent to the very high-to-critical pressure degree, and the lowest pressure, or very low-to-low level in District 15. According to the group of indicators of this component, it can be concluded that District 6 had a very high-to-critical pressure degree or critical level, 9.09% of the districts (Districts 3 and 12) had a high-to-very high degree or threshold level, 18.18% of the districts (Districts 1, 2, 7, and 21) had a medium-to-high pressure, 63.64% of the districts (Districts 4, 5, 8, 9 to 11, 13, 14, 16 to 20, and 22) had a low-to-medium pressure degree, and finally, one district had a very low-to-low degree. According to Figure 7c (waste component), Districts 1, 3, 6 to 8, 10, 14, 16, and 17 had the highest pressure, equivalent to the very high-to-critical degree, and the western Districts 21 and 22 had the lowest pressure, equivalent to the low-to-medium degree. Based on the group of waste indicators, it can be concluded that 40.91% of the districts had very high-to-critical pressure levels equivalent to the critical degree. Of the districts, 27.27% had a high-to-very high pressure degree equivalent to the threshold level, 22.73% of the districts had a medium-to-high pressure degree, and 9.09% of them (Districts 21 and 22) had low-to-medium degrees.

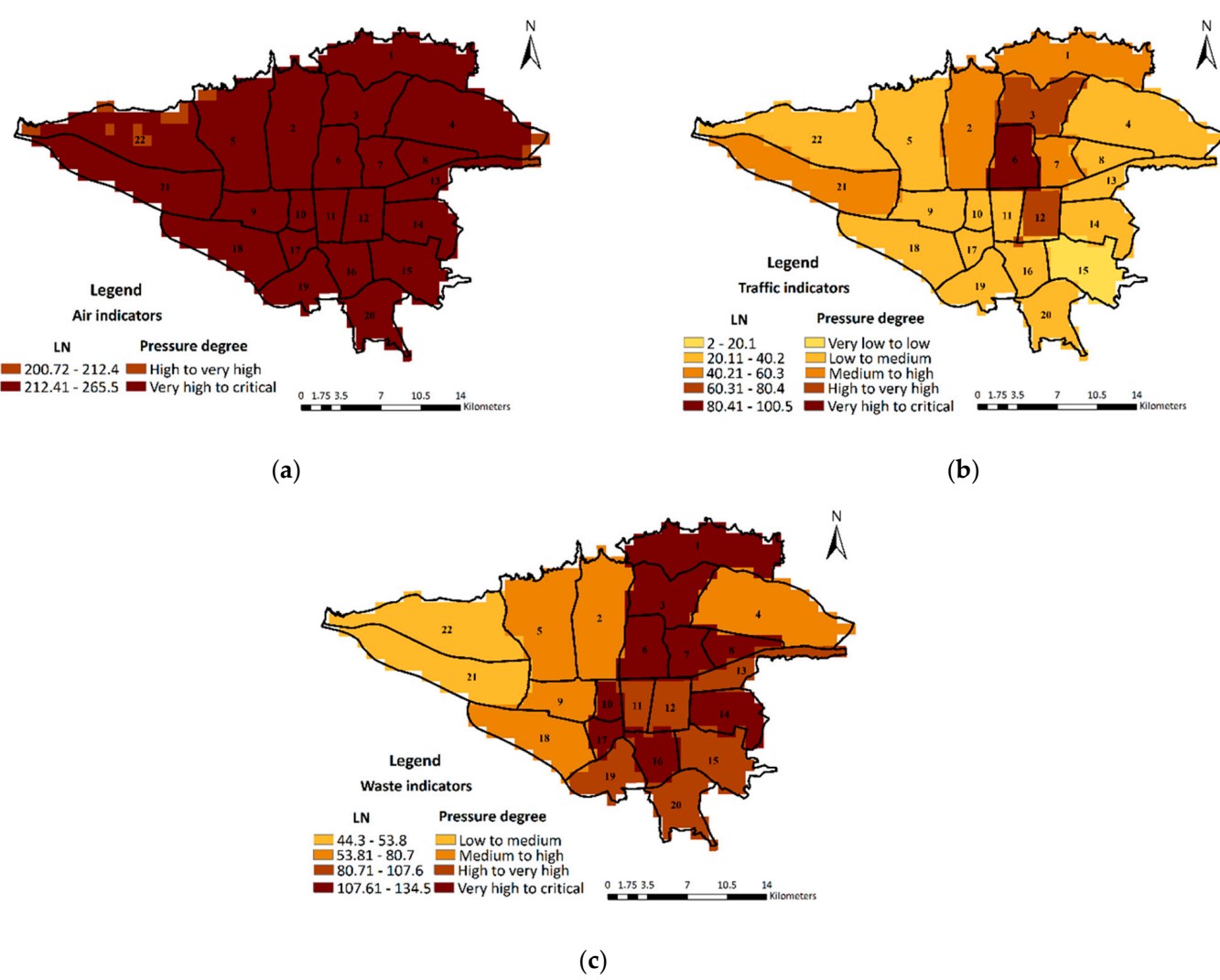

**Figure 7.** The LN maps of impact components. (**a**) Air; (**b**) traffic; (**c**) waste.

*3.5. Final LN Map of Human Impacts*

As mentioned, the LN map represents the environmental impact pressures resulting from human activities on the urban ecosystem. With the total load numbers of three components or 10 indicators, the LN map of human impacts was prepared at the spatial resolution of 1 km in Tehran, which is shown in Figure 8a. Additionally, this map averaged at the level of the districts depicted in Figure 8b. According to Figure 8a,b, Districts 1, 3, 6 to 8, 12, and 14 had very high-to-critical pressure degrees (critical UCC of impacts), and the rest of the districts, including Districts 2, 4, 5, 9 to 11, 13, and 15 to 22, had high-to-very high pressure levels (threshold UCC of impacts). The LN graph of human impacts in 22 districts of the study area is shown in Figure 9. From this chart, it can be observed that the total pressure on the ecosystem of Districts 6 (LN = 481.4), and 22 (LN = 306.6) were the highest and lowest, respectively, compared to other ones. Therefore, it can be mentioned that the environmental pressure degrees caused by the impacts of human activities in 31.82% of the districts (mainly in the northern, central, and eastern regions) were very high-to-critical, and 68.18% of the districts (mainly in the western and southern regions) were high-to-very high.

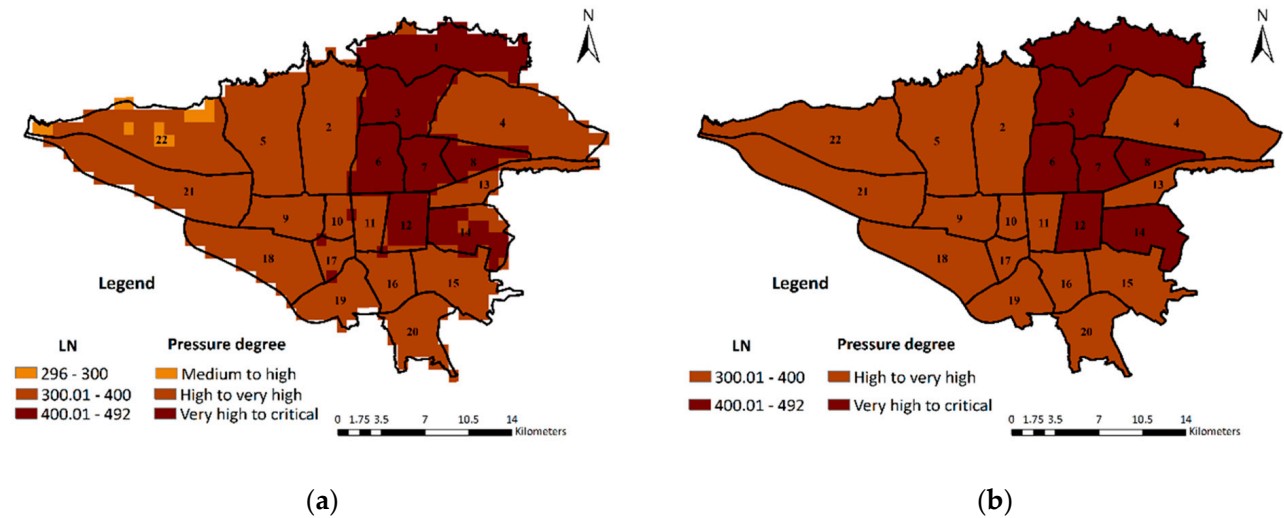

**Figure 8.** The LN map of human impacts. (**a**) The spatial resolution of 1 km; (**b**) level of 22 districts.

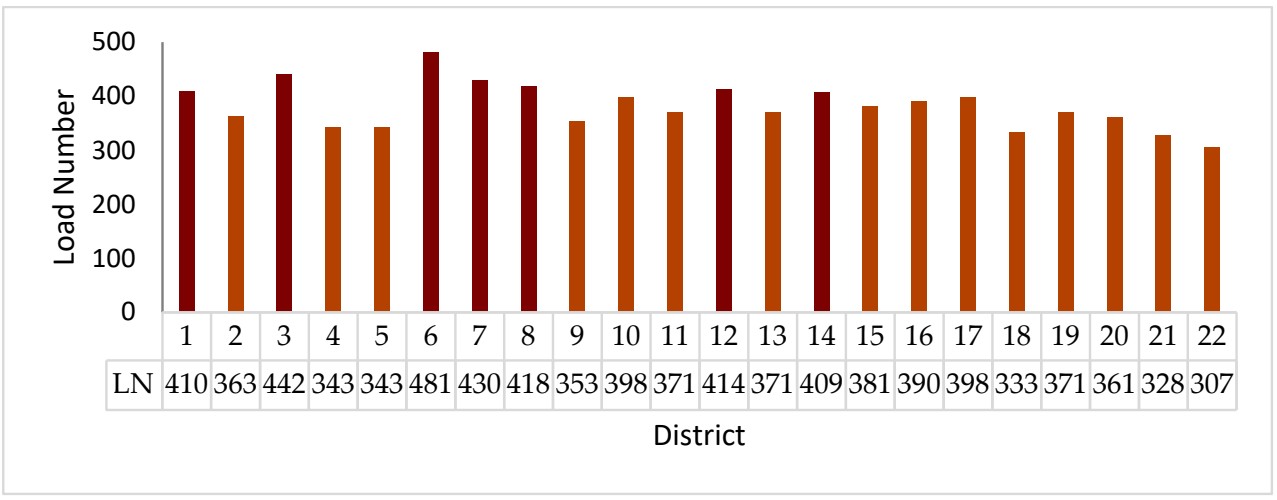

**Figure 9.** The LN graph of human impacts.

The ranking of the LN of the impacts of human activities on the environment of the districts in Tehran, in descending order, is presented in Table 4. From this table, Districts 6, 3, 7, 8, 12, 1, 14, 10, 17, 16, 15, 11, 13, 19, 2, 20, 9, 4, 5, 18, 21, and 22 had the highest to lowest pressure, respectively. Therefore, among the 22 districts of Tehran, western Districts, i.e., 22, 21, 18, and 5, had a more favorable situation, and northern, central, and eastern Districts, i.e., 6, 3, 7, 8, 12, 1, and 14, had a more unsuitable state according to air pollutants, traffic, and waste indicators. It can also be seen that all districts had high-to-critical pressure levels (LN = 306 to 481).

**Table 4.** Ranking of the LN of the human impacts on the ecosystem of each district of Tehran.

| Rank | District | LN | Pressure Degree | Rank | District | LN | Pressure Degree |
|------|----------|------|------------------|------|----------|------|------------------|
| 1 | 6 | 481.4 | very high-to-critical | 12 | 11 | 371.2 | high-to-very-high |
| 2 | 3 | 441.8 | | 13 | 13 | 371.0 | |
| 3 | 7 | 429.8 | | 14 | 19 | 370.6 | |
| 4 | 8 | 418.1 | | 15 | 2 | 363.1 | |
| 5 | 12 | 413.7 | | 16 | 20 | 361.4 | |
| 6 | 1 | 409.9 | | 17 | 9 | 353.3 | |
| 7 | 14 | 408.5 | | 18 | 4 | 342.9 | |
| 8 | 10 | 397.9 | high-to-very-high | 19 | 5 | 342.9 | |
| 9 | 17 | 397.9 | | 20 | 18 | 333.1 | |
| 10 | 16 | 390.1 | | 21 | 21 | 327.9 | |
| 11 | 15 | 380.9 | | 22 | 22 | 306.6 | |

## 4. Discussion

In this study, the LN map of the air component showed that the environmental pressure degree was very high-to-critical in all 22 districts. According to the Tehran Air Quality Control Company report [28], Tehran had only 17 clean days in 2020. Salehibarmi et al. [18] concluded in their study that the air quality in the majority of the 22 districts of Tehran was very poor, and Shabani et al. [19] also mentioned that the concentration of particulate matter (PM) in all air quality monitoring stations was more than the maximum acceptable standard, which makes their results similar to the LN maps of air indicators in this research. Gharehbakhsh et al. [15] also indicated that air pollution was more unfavorable in the central and southern districts of Tehran, which was in line with the results of this study that showed the critical pressure in the central districts in terms of $NO_2$, and in the southern districts in terms of $SO_2$. Moreover, due to more urban traffic jams in the central districts, higher $NO_2$ emissions were expected in these districts, and due to the greater distribution of industries and factories in the southern regions, higher $SO_2$ emissions were expected in these mentioned regions, too. Nababaksh et al. [16] and Shabani et al. [19] concluded that the green space per capita in the central and southern districts of Tehran was less favorable. These districts must have greater air pollution pressure, which was similar to the findings of this research.

The LN map of the traffic component in the current research represented the critical and very low-to-low pressure degrees in Districts 6 and 15, which had the highest, and the lowest trip generation and attraction rates, respectively. The pressure map of this component can be verified based on the $NO_2$ pollutant distribution LN map, because this pollutant is mainly concentrated in road traffic transportation areas due to the increased emission of fossil fuels. The LN of this pollutant in the central Districts, i.e., 3, 6, 7, and 12, was higher and close to critical, where the pressure of the traffic component was concluded to be medium-to-critical degree.

The LN map of the waste component showed very high-to-critical pressure degrees in Districts 1, 3, 6 to 8, 10, 14, 16, and 17, and low-to-medium pressure in the western Districts, i.e., 21 and 22. The population density in Districts 21 and 22 was 38 and 39 persons/ha,

respectively [26], and therefore, it is logical that the values of LN of the waste indicators have been much lower in these western districts compared to others.

Based on the final LN map of human impacts, it can be concluded that the northern, central, and eastern districts, including Districts 1, 3, 6 to 8, 12, and 14, had a more unfavorable environmental pressure situation (very high-to-critical degree), whereas western districts, including 5, 18, 21, and 22, had a more favorable state than others (Table 4). Alavi et al. [14] also found western Districts 21 and 22 to be the most stable regions in terms of environmental conditions. Moreover, the findings of Shabani et al. [19] indicated that compliance with environmental regulations in Districts 21 and 22 was better than in other regions. In the research of Esfandi and Nourian [22], the carrying capacity values of the western districts of Tehran were estimated to be acceptable, and the majority of the districts were estimated to be poor, and very poor. Further, Gharehbakhsh et al. [15] mentioned in their study that Tehran suffers from various types of pollution and environmental pressures due to the high population density. Their results were consistent with the findings of this study that none of the Tehran districts had desirable-to-high environmental pressure degrees in 2020. Therefore, in general, the results of all mentioned studies were consistent with the findings of this research.

Azizi [21] determined the criteria of air pollution based on the Air Quality Index (AQI) and the number of clean days adopted by the Tehran Air Quality Control Company to estimate the environmental carrying capacity. Furthermore, Esfandi and Nourian [22] and Abbaszadeh Tehrani and Makhdoum [5] demonstrated that one of the indicators used in estimating the environmental carrying capacity was the air pollution index obtained from the air quality monitoring stations. The advantage of the EUCCLN model compared to these studies is the use of remote sensing products of $NO_2$ and $SO_2$ pollutants in an appropriate spatial resolution. In Azizi [21], modeling was based on the weighted sum of the estimated importance coefficients of the indicators in their normalized values. Another advantage of the EUCCLN model against this research was the classification of each indicator based on the defined DCC table in estimating the environmental pressures caused by human impacts on the ecosystem.

The limitation of this research was not considering the relatively long time of the study period to determine the minimum and maximum values of $NO_2$, $SO_2$, and trip generation and attraction rate indicators to specify the desirable and critical values in their related DCC tables. Therefore, to overcome this limitation, it is suggested that in future studies, the DCCs of the mentioned indicators be determined based on their values over a longer period of time. It is also recommended to estimate the impacts of human activities in monthly or seasonal intervals to achieve more specific results. It is also suggested that the subjective weighting of indicators and ranking of alternatives be done under a fuzzy context.

## 5. Conclusions

In recent decades, due to the high population density, the city of Tehran has suffered from many environmental problems, such as the increase in air pollution, urban traffic, and excessive waste production caused by rapid development, without considering an efficient plan, and as a result, a lot of environmental pressures have been placed on its human and natural systems. The main aim of this research was to monitor the environmental pressures caused by the impacts of human activities (the third part of the PSIR framework) on the ecosystem of Tehran in 2020. For this purpose, the three components of air, traffic, and waste consist of a total of 10 indicators of the average concentration of $NO_2$, $SO_2$, $PM_{2.5}$, $PM_{10}$, and $O_3$ pollutants (air component), intra-city trip generation rate, and intra-city trip attraction rate (traffic component), and waste production, waste production rate, and recycling rate (waste component) were considered based on the concepts of the UCC, and sustainability. In Tehran, according to the $NO_2$ and $SO_2$ LN maps, the central and southern districts had higher pressure than other regions, respectively, and according to the $PM_{2.5}$, $PM_{10}$, and $O_3$, all of the districts had critical pressure degrees. Therefore, it was concluded that the air component in all 22 districts had a critical UCC limit. Based on

the LN map of traffic indicators, it was observed that the central and northern districts had higher pressure than other areas. According to the LN map of the waste production indicator, the central and southeastern districts had higher pressure, according to the LN map of the waste production rate, Districts 1, 3, 6, 8, 10, 14, and 17 had critical pressure degrees, and based on the LN map of the recycling rate, all 22 districts had critical level. Therefore, the component of waste in the northern, central, and southern districts, with higher pressures than other areas, was obtained. Based on the distribution LN map of total impacts, it was concluded that the human-induce environmental pressures in 31.82% of the districts (mainly northern, central, and eastern Districts 1, 3, 6 to 8, 12, and 14), were very high-to-critical, equivalent to critical UCC, and in 68.18% of the remaining districts (mainly western, and southern Districts 2, 4, 5, 9, 10, 11, 13, and 15 to 22) were high-to-very high, equivalent to the threshold degree. The overall environmental pressure of humans on the ecosystem of all districts has been several times higher than the desirable level, which has either reached or exceeded its maximum acceptable threshold. Thus, a further increase in impact pressures may cause irreversible change or permanent damage to its environment. The results of this research showed that some efficient strategies should be considered in the policies and plans of Tehran to reduce human-induced pressures and sustain living within the natural limits of the city. These findings can be used by environmental experts, decision-makers, and city managers to make optimal decisions simply and expressively, to improve the environmental conditions of the urban ecosystem.

**Author Contributions:** Conceptualization, Nadia Abbaszadeh Tehrani and Milad Janalipour; methodology, Nadia Abbaszadeh Tehrani, Farinaz Farhanj and Milad Janalipour; software, Nadia Abbaszadeh Tehrani and Farinaz Farhanj; validation, Nadia Abbaszadeh Tehrani, Farinaz Farhanj and Milad Janalipour; formal analysis, Nadia Abbaszadeh Tehrani, Farinaz Farhanj and Milad Janalipour; investigation, Nadia Abbaszadeh Tehrani and Farinaz Farhanj; resources, Farinaz Farhanj; data curation, Nadia Abbaszadeh Tehrani, Farinaz Farhanj and Milad Janalipour; writing—original draft preparation, Farinaz Farhanj; writing—review and editing, Nadia Abbaszadeh Tehrani, Farinaz Farhanj and Milad Janalipour; visualization, Farinaz Farhanj; supervision, Nadia Abbaszadeh Tehrani and Milad Janalipour; project administration, Nadia Abbaszadeh Tehrani; funding acquisition, Nadia Abbaszadeh Tehrani. All authors have read and agreed to the published version of the manuscript.

**Funding:** This research was funded by the Tehran Urban Research and Planning Center, Tehran Municipality, contract number 137/583608.

**Data Availability Statement:** Publicly available datasets analyzed in this study were collected from Sentinel-5P Nitrogen Dioxide (https://developers.google.com/earth-engine/datasets/catalog/COPERNICUS_S5P_NRTI_L3_NO2, accessed on 22 February 2022), Sentinel-5P Sulphur Dioxide (https://developers.google.com/earth-engine/datasets/catalog/COPERNICUS_S5P_NRTI_L3_SO2, accessed on 22 February 2022), Tehran Air Quality Company (https://airnow.tehran.ir/home/dataarchive.aspx, accessed on 13 April 2022), Tehran Municipality ICT Organization (https://tmicto.tehran.ir/, accessed on 12 February 2022), and the article by Jamialahmadi et al. (2020) (https://link.springer.com/article/10.1007/s10163-022-01423-8, accessed on 20 September 2022).

**Acknowledgments:** This work is derived from a part of a research project carried out for the Tehran Urban Research and Planning Center, Tehran Municipality, under the title: Estimation of environmental pressure on Tehran city using the urban carrying capacity load number model (UCCLN). The authors express their gratitude to the Deputy of Urban Environment Infrastructure and Innovation of the aforementioned center for effective cooperation and interaction with the authors, as well as the European Space Agency, the Tehran Air Quality Control Company, and the Tehran Municipality ICT Organization for providing the data and information. The authors would like to thank the "Environmental Remote Sensing" Research Laboratory of the Aerospace Research Institute (ARI), Ministry of Science, Research and Technology of Iran (MSRT) for supporting this study.

**Conflicts of Interest:** The authors declare no conflict of interest.

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
