# Peer review of "Monitoring the Impacts of Human Activities on Urban Ecosystems Based on the Enhanced UCCLN (EUCCLN) Model"

_ijgi, doi:10.3390/ijgi12040170_

Round 1
Reviewer 1 Report
In the paper ”Monitoring the impacts of human activities on urban ecosystem based on the Enhanced UCCLN (EUCCLN) model”, the authors investigate the the environmental pressures caused by the impacts of human activities on 14 the ecosystem of Tehran city using spatial indicators.
The ”Abstract” and ”Introduction” sections briefly describe and introduce us to the purpose and context of the research presented in this paper. Research methods are coherently described referring to: study area, data, methodology.
The results are presented according to the analyzed factors: importance Coefficient (IC) of Indicators; the DCC Table of Indicators; DCC and LN Maps of Indicators; LN Maps of Components; final LN Map of Human Impacts. The results are well coordinated in relation to the research methods.
The discussions and conclusions are relevant in the context of the results obtained.
Author Response
Comments from Reviewer 1
General Comment: In the paper ”Monitoring the impacts of human activities on urban ecosystem based on the Enhanced UCCLN (EUCCLN) model”, the authors investigate the environmental pressures caused by the impacts of human activities on 14 the ecosystem of Tehran city using spatial indicators.
The ”Abstract” and ”Introduction” sections briefly describe and introduce us to the purpose and context of the research presented in this paper. Research methods are coherently described referring to: study area, data, and methodology.
The results are presented according to the analyzed factors: importance Coefficient (IC) of Indicators; the DCC Table of Indicators; DCC and LN Maps of Indicators; LN Maps of Components; final LN Map of Human Impacts. The results are well coordinated in relation to the research methods.
The discussions and conclusions are relevant in the context of the results obtained.
Response: Thank you very much for agreeing with us on the intention of this manuscript.
Reviewer 2 Report
This study constructs a pathway for measuring the pressures generated by human activities on urban ecosystems by improving the indicators and analytical platform of the EUCCLN model. Tehran was used as an example for practical application and noteworthy results were obtained. In the context of increasing ecological pressures caused by urbanization, the topic of this manuscript is of practical significance and falls within the scope of this journal. However, there are several key points that require further explanation or revision by the authors:
(1) It is recommended to include the EUCCLN model in the keywords as it is one of the core contents of this research.
(2) More details about the AHP method are needed, such as the structure of the questionnaire and the number of participating experts. These information are crucial to ensure the validity of the evaluation outcomes.
(3) Why were the DCCs for NO2, SO2, and traffic indicators determined based only on current situations? Does this mean that the current highest value of these indicators represents the upper limit of the city’s carrying capacity? The determination of DDC may need to take into account differences in environmental capacity in the study area. For example, the environmental capacity of air pollutants may be related to the local wind environment.
(4) The RESULTS section provides a comprehensive presentation of the research findings, but there is some repetition between the RESULTS and DISCUSSION sections. It is recommended that the authors add a comparison with previous studies in the DISCUSSION section, such as the advantages of the EUCCLN model - especially in the improvement of geographic information platforms - and differences in environmental pressures caused by urban development or other reasons.
Additionally, it is recommended that the authors optimize their English expressions.
Author Response
Comments from Reviewer 2
- General Comment: This study constructs a pathway for measuring the pressures generated by human activities on urban ecosystems by improving the indicators and analytical platform of the EUCCLN model. Tehran was used as an example for practical application and noteworthy results were obtained. In the context of increasing ecological pressures caused by urbanization, the topic of this manuscript is of practical significance and falls within the scope of this journal. However, there are several key points that require further explanation or revision by the authors:
Response: Thank you very much. We have read your comments carefully and tried our best to address them one by one. We hope that the manuscript has
been improved after this revision.
- Comment 1: It is recommended to include the EUCCLN model in the keywords as it is one of the core contents of this research.
Response: Thank you very much for pointing this out. We added the Enhanced Urban Carrying Capacity Load Number (EUCCLN) model into the keywords section [Pg1].
- Comment 2: More details about the AHP method are needed, such as the structure of the questionnaire and the number of participating experts. These information are crucial to ensure the validity of the evaluation outcomes.
Response: Thank you very much for the reminder. We added relevant explanations as follows: “Analytical Hierarchy Process (AHP) [41], which is one of the most popular methods of Multi-Criteria Evaluation (MCE), was used to determine the Importance Coefficient (IC) of each of the 10 indicators in estimating the UCC of impacts. AHP is a method for making decisions in complex problems that include a large number of indicators and the importance of each of them needs to be determined. The weights of indicators were carried out by three experts in environmental and urban sciences in the Expert Choice software. The experts were familiar with the environmental problems of Tehran’s ecosystem and the concepts of UCC and sustainability. Moreover, the determined weights were confirmed by experts of the Tehran municipality. The structure of our decision making problem was designed in two levels including criteria and sub-criteria. Four questionnaires based on Saaty’s pairwise comparison table were employed. The first questionnaire is regarding the criteria i.e., air, traffic, and waste. The second, third, and fourth questionnaires are used for the sub-criteria, i.e., (NO2, SO2, PM2.5, PM10, O3), (generation rate, attraction rate), and (waste production, waste production rate, recycling rate).” [Pg6].
- Comment 3: Why were the DCCs for NO2, SO2, and traffic indicators determined based only on current situations? Does this mean that the current highest value of these indicators represents the upper limit of the city’s carrying capacity? The determination of DDC may need to take into account differences in environmental capacity in the study area. For example, the environmental capacity of air pollutants may be related to the local wind environment.
Response: Thank you. It is a good question. The determination of the DCC of the average concentration of NO2, SO2, and trip generation and attraction rate was done according to the method adopted in the study of Esfandi and Nourian (2021) (Esfandi, S.; Nourian, F. Urban carrying capacity assessment framework for mega mall development. A case study of Tehran’s 22 municipal districts. Land use policy. 2021, 109, 105628.) based on their minimum to maximum values. Since there is no standard for the range of remote sensing products (since they are new products), it is necessary to define this standard over a longer period of time. Therefore, we accept that it is possible to improve the quality of the study by defining a new standard. To overcome this limitation, it is suggested that in future studies, the DCCs of the mentioned indicators be determined based on their values over a longer period of time [Pg14].
- Comment 4: The RESULTS section provides a comprehensive presentation of the research findings, but there is some repetition between the RESULTS and DISCUSSION sections. It is recommended that the authors add a comparison with previous studies in the DISCUSSION section, such as the advantages of the EUCCLN model - especially in the improvement of geographic information platforms - and differences in environmental pressures caused by urban development or other reasons.
Response: Thanks for your suggestion. We added the following paragraph to the Discussion section. “Azizi [21] determined the criteria of air pollution based on the Air Quality Index (AQI) and the number of clean days adopted by the Tehran Air Quality Control Company to estimate the environmental carrying capacity. Also, In Esfandi and Nourian [22] and Abbaszadeh Tehrani and Makhdoum [5], one of the indicators used in estimating the environmental carrying capacity was the air pollution index obtained from the air quality monitoring stations. The advantage of the EUCCLN model compared to these studies is the use of remote sensing products of NO2 and SO2 pollutants in an appropriate spatial resolution. In Azizi [21] modeling was based on the weighted sum of the estimated importance coefficients of the indicators in their normalized values. Another advantage of the EUCCLN model against this research was the classification of each indicator based on the defined DCC table in estimating the environmental pressures caused by human impacts on the ecosystem.”[Pg14]. In addition, the advantages of the EUCCLN model compared to its initial version were also presented in [Pg3-4].
- General Comment: Additionally, it is recommended that the authors optimize their English expressions.
Response: We went through the entire manuscript to optimize English expressions.
Reviewer 3 Report
This paper monitors the impacts of human activities on urban ecosystem based on an enhanced urban carrying capacity load number. The paper is well-written although the innovative is little. My observations are as follows:
1. The objectives of the study should be explicitly stated
2. It's unclear to me how the data were preprocessed. While one of the innovation introduced in this work as claimed by the authors was the use of remote sensing data, the authors failed to explain the sources of the data, spatial resolution, the algorithms that were used to obtain the products, the accuracy and the reliability of the satellite data products.
3. The discussion section needs to draw much from the current literature to compare result, suggest further work and limitations.
4. I suggest that the authors should reflect on their chosen approaches and data for them to discuss uncertainty and limitations of their work. I can't find any of such discussion through out the manuscript.
Author Response
Comments from Reviewer 3
- General Comment: This paper monitors the impacts of human activities on urban ecosystem based on an enhanced urban carrying capacity load number. The paper is well-written although the innovative is little. My observations are as follows:
Response: Thank you very much for your comments that helped us improve this manuscript.
- Comment 1: The objectives of the study should be explicitly stated.
Response: Thank you for your nice reminder. The main purpose of the research was mentioned in [Pg2] (“The main purpose of this research was to use an Enhanced UCCLN (EUCCLN) model to monitor the environmental pressures caused by the negative impacts of human activities (the third part of the PSIR framework) on the urban ecosystem of Tehran city”). Specific objectives were also explicitly added as follows: “Also, the specific objectives of this research were to estimate the distance of the spatial indicators of human impacts with desirable, acceptable, threshold, and critical values based on the concepts of UCC and the integration of remote sensing and Geographic Information System (GIS) for better monitoring of urban ecosystem.” [Pg2-3].
- Comment 2: It's unclear to me how the data were preprocessed. While one of the innovation introduced in this work as claimed by the authors was the use of remote sensing data, the authors failed to explain the sources of the data, spatial resolution, the algorithms that were used to obtain the products, the accuracy and the reliability of the satellite data products.
Response: Thanks for your comment. NO2 and SO2 pollutants were collected using remote sensing products. The source of their collection was the Sentinel 5-P satellite, Tropospheric Monitoring Instrument (TROPOMI) sensor. Explanations related to this satellite, such as launch year, design goals, spatial resolution, and the bands, were presented in [Pg4] of the manuscript. As mentioned on this page, these images are taken from Google Earth Engine (GEE). Other explanations related to your comment were also added in [Pg4] as follows: “The average values ​​of mentioned pollutants (mol/m2) were obtained from the GEE in 2020 for the city of Tehran in GeoTIFF format. GEE is a high-performance open-source cloud computing system for storing, processing, visualizing, and analyzing time series of geospatial data and remote sensing products. It should be noted that NO2_column_number_density and SO2_column_number_density bands were used in this regard. Some researchers have concluded in their studies that the accuracy of Sentinel 5-P in monitoring the mentioned pollutants ranges from 0.5 to 0.81. Veefkind et al. have also indicated that the accuracy and reliability of Sentinel 5-P in monitoring pollutants is acceptable and meets the needs.” [Pg4].
- Comment 3: The discussion section needs to draw much from the current literature to compare result, suggest further work and limitations.
Response: Thanks for your suggestion. The recommendations for future studies were moved from the Conclusion section to the Discussion section [Pg14]. Also, limitations were added to the Discussion section as follows: “The limitation of this research was not considering the relatively long time of the study period to determine the minimum and maximum values of NO2, SO2, and trip generation and attraction rate indicators to specify the desirable and critical values in their related DCC tables. Therefore, to overcome this limitation, it is suggested that in future studies, the DCCs of the mentioned indicators be determined based on their values over a longer period of time. It is also recommended to estimate the impacts of human activities in monthly or seasonal intervals to achieve more specific results. It is also suggested that subjective weighting of indicators and ranking of alternatives be done under a fuzzy context.” [Pg14].
- Comment 4: I suggest that the authors should reflect on their chosen approaches and data for them to discuss uncertainty and limitations of their work. I can't find any of such discussion through out the manuscript.
Response: Thank you for your nice reminder. Limitations were added to the Discussion section as follows: “The limitation of this research was not considering the relatively long time of the study period to determine the minimum and maximum values of NO2, SO2, and trip generation and attraction rate indicators to specify the desirable and critical values in their related DCC tables.” [Pg14].
Round 2
Reviewer 2 Report
The authors have responded to all my comments and made revisions to most of them. The topic of this manuscript falls within the scope of this journal, the structure is clear, and the results and conclusions are reasonable. It can be received in its present form.
Reviewer 3 Report
I'm happy with your revisions